# Adverse Effects of Oral Cannabidiol: An Updated Systematic Review of Randomized Controlled Trials (2020–2022)

**DOI:** 10.3390/pharmaceutics14122598

**Published:** 2022-11-25

**Authors:** José Diogo R. Souza, Julia Cozar Pacheco, Giordano Novak Rossi, Bruno O. de-Paulo, Antonio W. Zuardi, Francisco S. Guimarães, Jaime E. C. Hallak, José Alexandre Crippa, Rafael G. Dos Santos

**Affiliations:** 1Department of Neurosciences and Behavior, Ribeirão Preto Medical School, University of São Paulo, Ribeirão Preto 14049-900, Brazil; 2National Institute for Science and Technology-Translational Medicine, Ribeirão Preto 14049-900, Brazil; 3Department of Pharmacology, Ribeirão Preto Medical School, University of São Paulo, Ribeirão Preto 14049-900, Brazil

**Keywords:** cannabidiol, CBD, safety, adverse effects, drug interaction

## Abstract

(1) Background: With the massive demand for the use and commercialization of medicinal cannabidiol (CBD) products, new randomized clinical trials (RCTs) are being published worldwide, with a constant need for safety and efficacy evaluation. (2) Methods: We performed an update on a systematic review published in 2020 that focused on analyzing the serious adverse effects (SAEs) of CBD in RCTs and its possible association with drug interactions. We also updated the report of the most prevalent CBD adverse effects (AEs). We systematically searched EMBASE, MEDLINE/PubMed, and Web of Science without language restriction for RCTs that reported adverse effects after repeated oral CBD administration for at least one week in healthy volunteers or clinical samples published from January 2019 to May 2022. The included studies were assessed for methodological quality by the Quality Assessment of Controlled Intervention Studies tool. The present review is registered on PROSPERO, number CRD42022334399. (3) Results: Twelve studies involving 745 randomized subjects analyzed were included (range 1.1–56.8 y). A total of 454 participants used CBD in the trials. The most common AEs of CBD were mild or moderate and included gastrointestinal symptoms (59.5%), somnolence (16.7%), loss of appetite (16.5%), and hypertransaminasemia (ALT/AST) (12.8%). Serious adverse effects include mainly hypertransaminasemia with serum levels elevations greater than three times the upper limit of the normal (6.4%), seizures (1.3%), and rash (1.1%). All SAEs reported in the studies were observed on CBD as an add-on therapy to anticonvulsant medications, including clobazam and valproate. (4) Conclusion: Recent RCTs involving oral CBD administration for at least a week suggest that CBD has a good safety and tolerability profile, confirming previous data. However, it can potentially interact with other drugs and its use should be monitored, especially at the beginning of treatment.

## 1. Introduction

Cannabis has been used for centuries for medicinal purposes. In the last two decades, decriminalization policies and new scientific evidence have significantly increased interest in the therapeutic potential of cannabis and its derivatives [1]. As a result, in several countries, there was a continuous process of authorizations for the marketing of cannabis-based products (CBMPs). CBMPs can range from purified single compounds, commonly cannabidiol (CBD) or delta-9-tetrahydrocannabinol (THC), to complex mixtures of hundreds of components in multiple formulations (e.g., oils, solutions, capsules, sublingual sprays), various routes of administration (e.g., oral, inhalation, topical), and differ in their manufacturing processes and quality control [2].

Cannabidiol (CBD) is a non-psychotomimetic phytocannabinoid with potential therapeutic properties across diverse physical and neuropsychiatric conditions due to its anti-inflammatory, analgesic, neuroprotective, anticonvulsant, and anxiolytic actions, among others [3]. There is increasing social pressure in many countries for patients to legally use this cannabinoid to treat anxiety, depression, sleep disorders, pain, and more extensively, promote quality of life and well-being [4,5]. However, in the United States and Europe, only the oral purified CBD preparation Epidiolex^®^ (GW-Pharm, United Kingdom) has been licensed to treat seizures associated with Lennox-Gastaut syndrome, Dravet syndrome, and Tuberous Sclerosis Complex. Some of the most common side effects of Epidiolex^®^ include sleepiness, decreased appetite, diarrhea, increased liver enzymes, sleep problems, and rash [6]. Thus, previous data suggest that oral cannabidiol is well tolerated and has relatively few serious adverse effects. Two recent systematic reviews [7,8] evaluating the safety of oral CBD in RCTs in healthy volunteers or clinical samples (a meta-analysis on general AEs and a systematic review focused on drug interactions and SAEs) reported AEs such as decreased appetite, diarrhea, somnolence, sedation, dizziness, headache, nausea, vomiting, abdominal discomfort, and rash. In addition, SAEs were elevated transaminases, convulsion, sedation, lethargy, and upper respiratory tract infections. For SAEs, both reviews reported being limited to childhood epilepsy studies that indicate that CBD may have interacted with other medications such as clobazam or sodium valproate.

Thus, with the increase in demand for the medicinal use of CBD, the emergence of new products, and recently published clinical studies, it is essential to follow the data on CBD effectiveness and its safety and tolerability. Therefore, the present review aims to address this issue by updating our previous systematic review on the SAEs of CBD and its interactions with other drugs in randomized clinical trials published between January 2019 to May 2022; we also extracted from the previous review and updated the report of the most prevalent AEs of CBD and evaluated the quality of the methodology used in the studies.

## 2. Methods

This systematic review was prospectively registered on PROSPERO (CRD42022334399) and reported according to the Preferred Reporting Items for Systematic Reviews and Meta-Analyses (PRISMA) statement. The PRISMA flowchart is shown in Figure 1. As an update of a systematic review, the methodology used was similar to the previous study [8], with minimal adaptations when necessary (see Appendix A).

### 2.1. Inclusion and Exclusion Criteria

We included double-blind, randomized controlled clinical trials that reported data on adverse effects from controlled trials using repeated oral administration in humans with formulations containing purified CBD (≥98% CBD). Open-label studies that did not include a placebo or drug comparison or lasted less than seven days were excluded. There were no restrictions concerning participant characteristics or disease indication.

### 2.2. Search Strategy and Study Eligibility

Randomized clinical trials on CBD published in peer-reviewed journals between January 2019 to May 2022 reported adverse effects identified using EMBASE, MEDLINE/PubMed, and Web of Science (independently performed by JS, JP, and GR); by screening the reference lists of articles identified and by correspondence with study investigators using the approach recommended by the PRISMA guidelines. The computer-based searches combined the words “(cannabidiol OR CBD) AND (randomized clinical trial OR double-blind OR placebo-controlled)” without language restriction.

### 2.3. Data Extraction and Quality Assessment

The following information was independently extracted from each article by three trained investigators (JS, JP, and GR) using a standardized form: study design, year of the publication, geographic location, sample size, the average age of participants, number and percentage of female participants, the condition being studied, cannabidiol treatment, control (placebo or other drugs), number and percentage of CBD users, CBD product formulation, safety assessment, adverse effects, serious adverse effects, and concomitant therapy. Adverse effects were classified into mild, moderate, and severe according to the reported categories. All discrepancies were resolved by consensus and, if necessary, by a research supervisor (RS) adjudication. The risk of bias and the methodological quality of the studies were assessed using the Quality Assessment of Controlled Intervention Studies tool from the National Heart, Lung, and Blood Institute [9]. Studies were considered methodologically poor, fair, or good based on the percentage of reported items < 50%, 50–75%, or >75%, respectively (see Appendix A); low risk of bias translates to a rating of good quality.

## 3. Results

### 3.1. Study Characteristics 

Twelve RCTs [10,11,12,13,14,15,16,17,18,19,20,21] involving 745 randomized subjects analyzed were included. The median number of participants per study was 62 (range 1.1–56.8 y). Eleven RCTs were placebo-controlled, and one was active-controlled [19]. A total of 454 participants used oral CBD in the trials, and 303 participants were female (40.7%). Treatment duration ranged from 1 to 16 weeks. Daily CBD doses included fixed doses (300 mg/day to 800 mg/day) and doses adjusted by weight (from 20 mg/kg/day to 50 mg/kg/day). All RCTs evaluated the effects of CBD on clinical populations and most used CBD concomitant with other drugs (9/12). Two studies evaluated the effects of cannabidiol on epilepsy, two on cocaine use disorder, one on psychosis, one on the high risk of psychosis, one on drug-resistant seizures in tuberous sclerosis complex, one on cannabis use disorder, one on COVID-19 patients, one on rapid eye movement sleep behavior disorder, one on behavioral problems in children and adolescents with intellectual disability, and one on functional dyspepsia. Three studies took place in Brazil, three in multiple countries, two in the United Kingdom, one in Australia, one in Canada, one in Germany, and one in the United States. All this information, in addition to concomitant therapy and investigational product formulations, are reported in Table 1.

### 3.2. Safety Assessment 

The safety of oral CBD isolate use was assessed differently across studies. Four studies evaluated laboratory parameters [11,12,15,21], two employed participants’ self-reports [13,18], two studies used the UKU Side Effects Scale [14,17], two did not apply a questionnaire or adverse event scale [10,19], one used the Systematic Assessment for Treatment Emergent Events (SAFTEE) tool [20], one used the Medical Dictionary for Regulatory Activities (MedDRA) [16], one used the Monitoring of Side Effects Scale (MOSES) [12], and one employed the Drug abuse liability [11].

### 3.3. Adverse Effects of CBD

CBD showed mild and moderate AEs in most studies (9/12). The most common AEs (≥10% incidence) included gastrointestinal symptoms (59.5%), somnolence (16.7%), loss of appetite (16.5%), increased ALT/AST (12.8%), and fatigue (11.4%). All AEs reported with an incidence ≥1% are described in Table 2. In most studies, participants used CBD concomitantly with another medication (see Table 1). Statistical analysis with Fisher’s exact test revealed significance (*p* ≤ 0.05) in using CBD for the increase in gastrointestinal symptoms, ALT/AST, rash, as well as change in appetite (both increasing and decreasing). In contrast, there was statistical significance for fever, with fewer cases for individuals who used CBD in the trials.

### 3.4. Serious Adverse Effects of CBD

Serious adverse effects of cannabidiol were reported in only three studies, all on epilepsy [11,15,16]. All SAEs reported in these studies were in the context of using CBD at doses between 20 mg/kg/day to 50 mg/kg/day as an add-on treatment to anticonvulsant medications, including clobazam and valproate. The most common SAEs were hypertransaminasemia (ALT/AST) with serum levels elevations greater than three times the upper limit (6.4%), seizures (1.3%), and rash (1.1%) (Table 3). Dosages greater than 25 mg/kg/day were associated with higher incidences of SAEs. For complete data on serious adverse effects (≥1 patient) see Appendix A. Statistical analysis with Fisher’s exact test revealed significance (*p* ≤ 0.05) in using CBD for ALT/AST increase.

### 3.5. Evaluation of the Studies’ Methodology and Bias Assessment

All studies showed an adequate to a good level of description. Half of the RCTs had fair methodological quality, while the other half had good quality. Thus, no study showed poor methodological quality. Two RCTs [13,17] reported 100% of items from the Quality Assessment of Controlled Intervention Studies tool (see Table 4), indicating a tendency towards a low bias risk.

## 4. Discussion

The present paper is an updated systematic review of the adverse effects of orally administered purified cannabidiol for different clinical conditions published in the last three years. The previous review [8] reported data from 18 RCTs published between 1980 and early 2020, involving 1127 subjects from clinical samples and healthy volunteers, with daily CBD treatment at doses ranging from 20 to 1500 mg/day and a duration ranging from one to 18 weeks. The results showed that CBD had mild and moderate adverse effects in most studies, the most common being drowsiness, sedation, fatigue, dizziness, headache, diarrhea, nausea, decreased appetite, and abdominal discomfort. Regarding SAEs, the highest incidence was reported in RCTs on epilepsy. Some of the most relevant effects were severe drowsiness, lethargy, hypertransaminasemia, rash, and pneumonia with or without respiratory failure. In the present review, we identified 12 trials with a total of 745 participants. In most studies, CBD showed mild and moderate adverse effects. The most common adverse effects were gastrointestinal symptoms (59.5%), which are probably related to the endocannabinoid system’s role in regulating gastrointestinal motility [22] and or by the product formulations (drug vehicle); other common AEs were somnolence (16.7%), loss of appetite (16.5%), increased ALT/AST (12.8%), and fatigue (11.4%). Although CBD was associated with a probability of SAEs, these events only occurred in epilepsy studies that used CBD as an add-on anticonvulsant treatment. SAEs with an incidence ≥1% were hypertransaminasemia (6.4%), seizures (1.3%), and rash (1.1%). Statistical analysis revealed a significant effect of CBD treatment for the increase in gastrointestinal symptoms, ALT/AST, rash, as well as change in appetite (both increasing and decreasing). In contrast, there was statistical significance for fever, with fewer cases for subjects who used CBD in the trials, possibly reflecting its anti-inflammatory effects. With the increase in the use of CBD in several countries and several RCTs being carried out, the strengths of our study included updating the evaluation of the most recent published studies and assessing their methodological qualities, resulting in a trend towards a low level of bias between the RCTs included. However, our analysis and conclusions are limited by the small sample size of the RCTs and the absence of studies with low doses of purified CBD.

Since cannabidiol is typically added to existing medicine treatments, interactions can occur between CBD and other co-administered medications. Pharmacokinetic drug-drug interactions can occur at the absorption, distribution, and elimination stages, resulting in changes in drug plasma concentrations. When cannabidiol is used with another medication, the pharmacokinetics of the CBD or the other drug may change [23]. CYP2C19 and, to a lesser extent, CYP3A4 are the main cytochrome P450 (CYP) enzymes that metabolize CBD, converting it to its primary active metabolite 7-hydroxy CBD [24,25]. These hepatic enzymes are also involved in the metabolism of several other drugs, including the anticonvulsants clobazam and valproate. CBD has been reported to inhibit CYP2C19 at doses as low as 5 mg/kg [26]. CYP2C19 inhibition increases the levels of N-desmethylclobazam, the main active metabolite in clobazam [26], and has the potential to interact with a broad range of other generally prescribed medications. The clinical impact of these interactions, particularly at the usual therapeutic doses of CBD, must be carefully investigated. Hypertransaminasemia has also been reported after CBD’s clinical use. Even if most cases occurred in epileptic patients using valproate, increases in hepatic transaminases were reported in open-label studies employing 300 mg/day CBD as a single medication [27]. The precise mechanism of valproate-CBD interaction is not fully understood. The co-administration of CBD and valproate does not significantly alter their plasma levels or metabolites [28]. However, 7-COOH-CBD, valproate and its metabolite 4-ene-valproic acid may affect hepatic mitochondrial function [26].

As SAEs, in addition to hypertransaminasemia, seizures and rash have also been reported, resulting in treatment discontinuation. The three studies reporting severe seizures were in the context of patients diagnosed with epilepsy, and it is not simple to differentiate the worsening of the baseline pathology (epilepsy) from an adverse effect of the treatment. Rash is one of the most common adverse drug reactions. It can present in a variety of ways, from mild rashes to extensive lesions with epidermal detachment associated with severe systemic involvement, as seen, for example, in Stevens-Johnson syndrome and toxic epidermal necrolysis. However, the rash is very non-specific and can often be found in other conditions, such as viral infections; in isolated CBD oil, the rash may be related to the medication vehicle, for example. Thus, rash diagnosis secondary to medication requires extensive laboratory investigation and histopathological confirmation. A case series [29] from an open-label clinical trial [30] involving CBD and rash were reported in four participants who were not using other medications. An extensive laboratory investigation, including histopathological analysis, was performed, in addition to re-exposure to the medication vehicle only, showing no new reactions, suggesting that the skin reactions were related to CBD.

While numerous countries have a growing demand for legal cannabidiol use to treat various mental and physical health problems, determining the best regulatory response to these requests is a substantial challenge. CBD products can come in a variety of formulations (e.g., purified CBD, CBD:THC ratios, CBD enriched products), routes of administration (e.g., oral [oil, tablets, capsules], sublingual spray, inhaled, topical) [2], and many are prepared without adequate manufacturing quality control, pharmaceutical grade production and labeling. There is a common perception that CBD is a “natural medicine”, even though its derivatives are commonly extracted from cannabis and synthetically modified. Furthermore, to date, there needs to be more robust data to support the indications of its therapeutic use for many of these conditions required by global demand. In some countries, CBD is marketed as a supplement, food, or pet product, which becomes a problem as no robust regulatory standards are required. Recently, the Science Advisory Committee of Health Canada reported some recommendations on Health Products Containing Cannabis—products containing cannabidiol can be purchased without needing a prescription from a medical professional, similar to other over-the-counter products [31]; At the same time, the European Food Safety Authority made a statement on the safety of cannabidiol as a novel food and, considering the significant uncertainties and data gaps, concluded that the safety of CBD as a novel food cannot currently be established [32].

With the increase in demand for the medicinal use of CBD, new high-quality studies are essential, with dose variations, particularly smaller doses, contemplating real-world studies with larger samples, and reporting adverse events, to better assess the effectiveness, safety, and tolerability of CBD. The clinical use of CBD must be aligned with many other pharmaceutical products, ensuring its effectiveness and safety for patients, and informing them of its potential adverse effects and interactions with other drugs. As new research emerges, it may become evident that there is a range of CBD doses at which clinically relevant adverse effects and drug interactions are likely to occur [23]. Therefore, as quality data is built, it is urgently necessary to inform and protect consumers, ensuring the quality and safety of CBD products.

## 5. Conclusions

The data from the present systematic review agree with previous data on the safety of purified CBD. The most common adverse effects are mild and moderate, and serious adverse effects are rare and have been only reported in epilepsy studies, with concomitant use of CBD with antiepileptic drugs. Physicians must carry out the indication for the use of CBD through prescription, and its use must be monitored, especially at the beginning of the treatment. Due to the high global demand for CBD in various conditions, additional safety data from clinical trials with larger samples, different dosages, and different products are needed (e.g., evaluating the effects in the comparison of purified CBD with broad-spectrum and full-spectrum formulations).

## Figures and Tables

**Figure 1 pharmaceutics-14-02598-f001:**
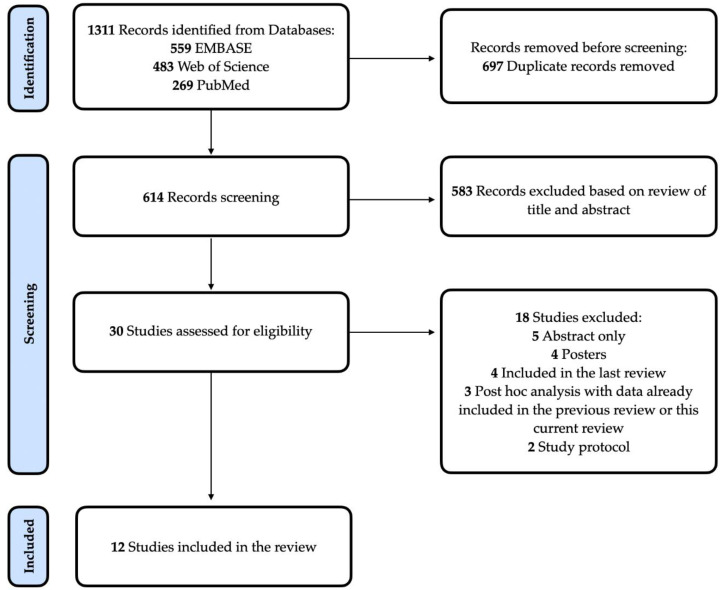
PRISMA Flow Diagram. Flowchart describing the systematic search strategy, including the identification, screening, and inclusion of relevant studies.

**Table 1 pharmaceutics-14-02598-t001:** A visual summary of the characteristics from included studies from the systematic review.

Reference	Study Design/Sample	Treatment	Investigational Product Formulation	Concomitant Therapy	Safety Assessment	Adverse Effects	Serious Adverse Effects
Appiah-Kusi et al., 2020 [10]	RCT, phase I, double-blind, between-groups, placebo-controlled. 33 clinical high-risk psychosis volunteers (Age 23–29 y) 16 CBD; 17 placebo 25 healthy volunteers (no use of intervention drug) (Age 23–29 y)	600 mg/day oral CBD for 1 week.	CBD capsules (STI Pharmaceuticals, UK).	None	No adverse event scale or questionnaire was applied	None	None
Ben-Menachem et al., 2020 [11]	RCT, phase II, two-arm, parallel-group, double-blind, placebo-controlled. 34 patients with epilepsy receiving Stiripentol (STP) or Valproate (VPA) and no more than 2 other AEDs (16–55 y). STP arm (n = 14): 12 CBD 2 placebo VPA arm (n = 20): 16 CBD 4 placebo	CBD 20 mg/kg/d administered as two equal doses twice a day for 2 weeks.	Epidiolex^®^ 100 mg/mL (GW Pharma, Cambridge, UK).	Carbamazepine Clobazam Clonazepam Ethosuximide Lacosamide Lamotrigine Levetiracetam Lorazepam Oxcarbazepine Rufinamide Stiripentol Topiramate VPA sodium Zonisamide	Laboratory parameters (liver enzymes) Drug abuse liability (monitored if CBD was overused or went missing)	STP arm: 8 CBD patients experienced mild AE (mostly diarrhea and fatigue). 2 CBD patients experienced ALT and AST increases (solved during the trial) VPA arm: 14 CBD patients experienced mil AE (mostly diarrhea) No AE in the placebo group.	STP arm: 1 generalized rash led to discontinuation. Rash solved after both CBD and STP discontinued. VPA arm: 1 hypertransaminasemia led to discontinuation. SAE solved by day 40.
Efron et al., 2020 [12]	RCT, phase I/II double-blind, parallel-group, placebo-controlled. 8 participants diagnosed with Intellectual Disability (8–16 y). 3 CBD 5 placebo	CBD was up titrated over 9 days from 5 mg/kg/day to 20 mg/kg/d in two divided doses, with a maximum dose of 500 mg twice/d for 8 weeks.	100 mg/mL CBD oral solution. 98% CBD in grapeseed oil (Tilray, Nanaimo, BC, Canada).	Clonidine Fluoxetine Guanfacine Melatonin Methylphenidate Risperidone Sodium Valproate	Safety: Blood collection (liver enzymes) Monitoring of Side Effects Scale (MOSES).	CBD: Eyes rolled up (1), tics (1), ear ringing (1), drooling (1), abdominal pain (1), decreased appetite (1), increased appetite (1), constipation (1), decreased wight (1), increased weight (1), restlessness (1), jitter (1), acne (1), urination incontinence (1), sadness (1), drowsiness (1), excessive sleep (1), insomnia (1). Placebo: Headache (1), nose congestion (1), increased appetite (1), increased weight (3), sadness (1), insomnia (1).	None
Freeman et al., 2020 [13]	RCT, phase II, double-blind, placebo-controlled, parallel-group. 48 patients in 1st stage plus 34 patients in 2nd stage with DSM-V criteria for a cannabis use disorder (at least moderate severity), expressing a desire to stop using cannabis (16–60 y). 77 participants completed treatment. 1st stage: 23 Placebo 12 CBD 200 mg 12 CBD 400 mg 12 CBD 800 mg 2nd stage: 23 Placebo 24 CBD 400 mg 23 CBD 800 mg	Gelatine capsules containing microcrystalline cellulose filler and CBD (50 mg, 100 mg, or 200 mg) for 4 weeks. 2 capsules twice daily: CBD 200 mg 4 capsules 50 mg CBD 400 mg 4 capsules 100 mg CBD 800 mg 4 capsules 200 mg	Synthetic CBD 99.9% purity (STI Pharmaceuticals, UK; manufactured by Nova Laboratories, UK).	None	Participants were asked about possible adverse events at each assessment from week 1 to week 16. All adverse events were verified with a medical supervisor and an independent trial monitor throughout the trial on an ongoing basis.	CBD 200 mg (12): 42 mild and 4 moderate AE CBD 400 mg (24): 96 mild and 8 moderate CBD 800 mg (23): 78 mild and 8 moderate Placebo (23): 65 mild and 9 moderate	None
Meneses-Gaya et al., 2020 [14]	RCT, double-blind, placebo-controlled. 31 patients with DSM-IV diagnosis of crack-cocaine dependence (>18 y). 14 CBD 17 placebo	CBD 300 mg/d. Two 150 mg capsules/d for 10 days.	CBD 99.9% pure powder (THC-Pharm, Germany/ STI-Pharm, Brentwood, UK) dissolved in corn oil.	Benzodiazepines	UKU Side Effects Rating Scale (UKU-SERS)	Sleepiness: 5 CBD; 3 placebo Nausea: 2 CBD; 1 placebo Headache: 2 CBD; 1 placebo	None
Thiele et al., 2020 [15]	RCT of add-on CBD vs. placebo, Phase III, double-blind, parallel-group. 224 included patients with a clinical diagnosis of Tuberous Sclerosis Complex (1–65 y). 75 CBD25 73 CBD50 76 placebo	CBD 25 mg/kg/day (CBD25) or 50 mg/kg/d (CBD50) for 16 weeks. 4 w for dose escalation (titration period) followed by 12 w of stable dosing (maintenance period).	Epidiolex^®^ 100 mg/mL (GW Pharma, UK).	Clobazam Valproic Acid	Safety was assessed primarily by evaluating adverse events and clinical laboratory parameters.	Most common: Diarrhea: 23 CBD25; 41 CBD50; 19 placebo. Somnolence: 10 CBD25; 19 CBD50; 7 placebo. Decreased appetite: 15 CBD25; 17 CBD50; 9 placebo. Liver transaminase level elevations: 17 CBD25; 30 CBD50	Serum aminotransferase level elevations greater than 3 times the upper limit of the normal range: 9 CBD25 19 CBD50 Rash 2 CBD25 2 CBD50 Seizure 3 CBD25 2 CBD50
VanLandingham et al., 2020 [16]	RCT, phase II, double-blind, placebo-controlled. 20 patients with epilepsy taking Clobazam and no more than 2 other antiepileptic drugs (18–65 y). Seven patients (1 taking placebo and 6 taking CBD) were excluded from the PK analysis	CBD 20 mg/kg/d coadministered with Clobazam. Patients titrated their CBD dose for 10 days (days 2 to 11) to 20 mg/kg/d CBD (twice daily). The titration period was followed by a 21d maintenance dose period (days 12 to 32).	Epidiolex^®^ 100 mg/mL (GW Pharma, UK).	Carbamazepine Clobazam Eslicarbazepine Lacosamide Lamotrigine Levetiracetam Oxcarbazepine Perampanel Phenobarbital Valproic acid	Medical Dictionary for Regulatory Activities (MedDRA)	Diarrhea: 6 CBD 6; placebo 1 Nausea: 3 CBD Vomiting: 3 CBD Dizziness: 2 CBD Sedation: 2 CBD Somnolence: 2 CBD Skin tissue disorders: 6 CBD	Seizure cluster: 1 CBD
Crippa et al., 2021 [17]	RCT, phase, II double-blind, parallel-group, placebo-controlled. 105 patients diagnosed with COVID-19 (18–65 y). The data of 91 patients were included in the final analysis: 49 CBD 42 placebo	CBD 300 mg/d administered as two equal doses twice for 2 weeks.	Oral CBD 150 mg/mL 99.6% purity (PurMed Global, USA).	Dipyrone Paracetamol	Modified UKU Side Effects Scale	Somnolence:38 CBD; 33 placebo Fatigue:38 CBD; 33 placebo Decreased appetite:38 CBD; 32 placebo Lethargy:25 CBD; 15 placebo Weight loss:24 CBD; 22 placebo Nausea:23 CBD; 16 placebo Diarrhea:21 CBD; 20 placebo Increased appetite:17 CBD; 10 placebo Fever:11 CBD; 15 placebo Weight gain:10 CBD; 8 placebo	None
De Almeida et al., 2021 [18]	RCT, phase II/III, double-blind, placebo-controlled, parallel-group. 33 patients with Parkinson’s Disease (>18y). 17 CBD 16 placebo	1st week 1 capsule (CBD 75mg) 2nd week 1 capsule (CBD 150 mg) the 3rd until the 12th w 2 capsules (CBD 150 mg)	CBD 99.6% pure powder form (BSPG Pharm, UK) dissolved in corn oil (capsule).	Antidepressants (SSRI or dual) Clonazepam Melatonin	Participants self-report	CBD: Epigastric pain (1), Nausea (1), headache (1), drowsiness (1), sadness (2), and dizziness (1). Placebo: Headache (1).	None
Leweke et al., 2021 [19]	RTC, phase II, parallel-group, active-controlled, mono-therapeutic, double-blind. 42 patients diagnosed with schizophrenia or schizophreniform psychosis (18–50 y). The data of 39 patients were included in the final analysis: 20 CBD 19 Amisulpride	Both CBD and AMI: Initial dose 200 mg/d and increased stepwise by 200 mg per day to a daily dose of 200 mg four times daily (800 mg/d) with the 1st w. Maintained for another 3 weeks (4w total).	Pharmaceutical grade not stated.	Lorazepam	No adverse event scale or questionnaire was applied	Although side effects have been reported (3 CBD; 5 AMI), they have not been described.	None
Mongeau-Pérusse et al., 2021 [20]	RCT: Phase II, double-blind, parallel-group, placebo-controlled. 78 diagnosed with current cocaine use disorder patients (18–65 y). Phase I: Detoxification (10 days); Phase II: 12-w outpatient follow-up. Completed Phase I: 35 CBD; 27 placebo Completed Phase II: 27 CBD; 23 placebo	CBD 300 mg/mL for 92 days. Days 2 and 3: CBD 400 mg (1.3 mL) and then increased the dose to 800 mg/day (2.7 mL) for the rest of the study.	Synthetic CBD 300 mg/mL (Insys Therapeutics, Phoenix, AZ, USA).	None	Systematic Assessment for Treatment Emergent Events (SAFTEE) tool	Diarrhea: 26 CBD; 1 placebo Nausea: 3 CBD; 3 placebo Abdominal pain: 3 CBD Hypoaesthesia: 2 CBD; 1 placebo Abdominal distension: 2 placebo Insomnia: 2 CBD	Placebo group: 1 Hepatitis
Atieh et al., 2022 [21]	RCT, double-blinded, placebo-controlled (1:1 ratio). 48 patients with Functional dyspepsia with normal gastric emptying (18–70 y) 25 CBD 23 placebo	CBD 20 mg/kg/d administered as two equal doses twice a day for 4 weeks.	Epidiolex^®^ 100 mg/mL (GW Pharma, UK).	Analgesic; antibiotic; anticonvulsant; antiemetic; anti-fungal; alpha-2 adrenergic; anxiolytic; antipsychotic; antispasmodic; anti-histaminic; anti-acid secretory agent; anti-migraine; anti-hypertensive; birth control hormones; birth control IUD; bronchodilator; dopaminergic/noradrenergic; epinephrine pen available; hormones; lipid reducing agent; night sedative; NSAIDs; SSRI; topical/nasal steroids; tricyclic antidepressant.	Laboratory parameters (liver enzymes)	Elevated liver enzymes: 4 CBD; 1 placebo Abdominal distension: 6 CBD; 2 placebo Nausea: 5 CBD; 1 placebo Headache: 3 CBD; 1 placebo Diarrhea: 7 CBD; 1 placebo Dizziness: 2 CBD Fatigue: 2 CBD; 3 placebo Loss of appetite: 2 CBD	None

AE adverse effect; AEDs antiepileptic drugs; AMI amisulpride; CBD cannabidiol; IUD intrauterine device; NSAID non-steroidal anti-inflammatory drug; RCT randomized controlled trial; SAE serious adverse effect; SSRI selective serotonin reuptake inhibitor; STP stiripentol; UKU acronym for the Danish name “Udvalg for Kliniske Undersøgelser”—Task force for clinical investigations; VPA valproate.

**Table 2 pharmaceutics-14-02598-t002:** The most common adverse effects reported in clinical trials on Cannabidiol.

Number of Subjects Showing Adverse Effects and Frequency (%)			
Adverse Effects	CBD	Control	Fisher’s Exact Test
Gastrointestinal symptoms	270 (59.5)	89 (30.6)	<0.0001
Diarrhea	145 (31.9)	42 (14.4)	<0.0001
Nausea	46 (10.1)	22 (7.6)	NS
Vomiting	35 (7.7)	11 (3.8)	0.029
Abdominal Pain	20 (4.4)	4 (1.4)	0.031
Constipation	14 (3.1)	6 (2.1)	NS
Abdominal Distention	10 (2.2)	4 (1.4)	NS
Somnolence	76 (16.7)	43 (14.8)	NS
Loss Appetite	75 (16.5)	32 (11.0)	0.041
ALT/AST Increase	58 (12.8)	1 (0.3)	<0.0001
Fatigue	52 (11.4)	40 (13.8)	NS
Increased Appetite	44 (9.7)	11 (3.8)	0.002
Headache	33 (7.8)	17 (5.8)	NS
Rash	29 (6.4)	2 (0.7)	<0.0001
Lethargy	26 (5.7)	15 (5.1)	NS
Weight Loss	25 (5.5)	22 (7.5)	NS
Nasopharyngitis	24 (5.3)	12 (4.1)	NS
Insomnia	15 (3.3)	5 (1.7)	NS
Upper respiratory tract infection	14 (3.1)	10 (3.4)	NS
Weight Gain	11 (2.4)	11 (3.8)	NS
Fever	11 (2.4)	21 (7.2)	0.002
Seizure	9 (2.0)	5 (1.7)	NS
Dizziness	6 (1.3)	0	NS

NS: non-significant (*p* > 0.05).

**Table 3 pharmaceutics-14-02598-t003:** The most common serious adverse effects reported in RCTs on CBD.

Number of Subjects Showing Serious Adverse Effects and Frequency (%)			
Serious Adverse Effects	CBD	Control	Fisher’s Exact Test
ALT/AST Increase	29 (6.4)	0	<0.0001
Seizure	6 (1.3)	1	NS
Rash	5 (1.1)	0	NS

NS: non-significant (*p* > 0.05).

**Table 4 pharmaceutics-14-02598-t004:** Summary of the analysis of the studies’ methodologies.

References	Items Where Information Was Present	Information Not Present or Not Reported	Total Applicable Items	Percentage of Reported Items	Quality Rating
Appiah-Kusi et al., 2020 [10]	9	5	14	64%	Fair
Ben-Menachem et al., 2020 [11]	10	4	14	71%	Fair
Efron et al., 2020 [12]	10	4	14	71%	Fair
Freeman et al., 2020 [13]	14	0	14	100%	Good
Meneses-Gaya et al., 2020 [14]	11	3	14	79%	Good
Thiele et al., 2020 [15]	9	5	14	64%	Fair
VanLandingham et al., 2020 [16]	7	7	14	50%	Fair
Crippa et al., 2021 [17]	14	0	14	100%	Good
De Almeida et al., 2021 [18]	11	3	14	79%	Good
Leweke et al., 2021 [19]	9	5	14	64%	Fair
Mongeau-Pérusse et al., 2021 [20]	11	3	14	79%	Good
Atieh et al., 2022 [21]	10	1	14	71%	Good

Quality Rating: Poor <50%, Fair 50–75%, Good >75% (See Appendix A).

## Data Availability

The data presented in this study are openly available in Appendix A.

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
