# Peer review of "Adverse Effects of Oral Cannabidiol: An Updated Systematic Review of Randomized Controlled Trials (2020–2022)"

_pharmaceutics, 2022, doi:10.3390/pharmaceutics14122598_

Round 1

Reviewer 1 Report

The proposed manuscript reviewed the most recent clinical trials on CBD used as the treatment of different pathologies published in the last three years, to assess the potential adverse effect of CBD and CBD pharmacological interactions with concomitant treatments. Since it represents the update of a previously published review, the data reported are limited. Nonetheless, the constant update of CBD toxicity is of crucial importance considering the growing interest in this molecule as a treatment. However, a brief description of the pathologies treated with CBD may provide a more complete overview of the current state of the art, including the research recently published in the literature (e.g.Brunetti P. et al. “Herbal Preparations of Medical Cannabis: A Vademecum for Prescribing Doctors.” Medicina (Kaunas, Lithuania) ). The applied methodology appears strong and well described. The first table should be improved since the reported information is not clearly provided. The investigated studies should be organized in chronological order, the column “concomitant therapy” should be moved near the Investigational “Product Formulation column” and the reported abbreviations should be specified as a footnote at the end of the table. Furthermore, all the references must be numbered and added to the reference list.

Taking into consideration that the review is presented as an update of the previous review, an extensive comparison to the former results should be provided and commented on in the “discussion section”.

Unfortunately, a complete review of the manuscript may not be provided due to an editing issue of the provided PDF which compromises the readability of pages 10,12, and 17-20.

Author Response

We revised the text accordingly, and incorporated the suggestions regarding the reference, table, and the comparison with the previous literature.

Reviewer 2 Report

- Is it possible to add the study duration of “Jan 2019-May 2022” in the Title?

- The introduction and Discussion sections seem to be a high similarity to the previously published works. I suggest paraphrasing the sentences to decrease the similarity index.

- The authors should clarify the “high-purity grade CBD”. How many limits of residual THC content can be accepted? Please mention, if possible.

- Please clarify CD, CR, and NA in the Supplementary file.

- Section 3.1 Study characteristics; Is it possible to add median or mode for age, treatment duration, and dose?

- Efron et al, 2020 used 98%CBD in oil. Is it can be classified as high-purity grade CBD?

- Table 2 should be rearranged.

- It will be nice if the author included statistical analysis with a p-value in Tables 2 and 3.

- Please add the reference for quality rating; poor, fair, and good classification in Table 4.

Author Response

- Is it possible to add the study duration of “Jan 2019-May 2022” in the Title?

We appreciate your comment and suggestion, and we incorporated information from the period of studies included in the review “(2020-2022)”.

- The introduction and Discussion sections seem to be a high similarity to the previously published works. I suggest paraphrasing the sentences to decrease the similarity index.

We agreed with the Referee's observation and revised the text accordingly.

- The authors should clarify the “high-purity grade CBD”. How many limits of residual THC content can be accepted? Please mention, if possible.

We welcome your comment and suggestion and have proofread the text. We added the description on line 63.

- Please clarify CD, CR, and NA in the Supplementary file.

We welcome your comments and suggestions and revised the text accordingly.

- Section 3.1 Study characteristics; Is it possible to add median or mode for age, treatment duration, and dose?

Data to calculate medians and modes are only available in some papers included. Therefore, we kept the means and ranges.

- Efron et al, 2020 used 98%CBD in oil. Is it can be classified as high-purity grade CBD?

We added the description on line 63.

- Table 2 should be rearranged.

The rearrangement was done.

- It will be nice if the author included statistical analysis with a p-value in Tables 2 and 3.

We agreed with the Referee's observation and included the p-value in Tables 2 and 3.

- Please add the reference for quality rating; poor, fair, and good classification in Table 4.

We have added the requested information on eTable 1, where the quality rating is fuly explainned.

Reviewer 3 Report

well presented summaries in 12 tables as well.

Author Response

This reviewer did not have any comments.

Round 2

Reviewer 1 Report

The can now be accepted in the opinion of the reviewer